# Secure Automatic Identification System (SecAIS): Proof-of-Concept Implementation

Athanasios Goudosis [1,2] and Sokratis Katsikas [3,*]

1   Systems Security Laboratory, Department of Digital Systems, University of Piraeus, 18532 Piraeus, Greece; a.goudosis@gmail.com
2   Hellenic Authority for Higher Education (HAHE), 10559 Athens, Greece
3   Department of Information Security and Communication Technology, Norwegian University of Science and Technology, 2802 Gjøvik, Norway
*   Correspondence: sokratis.katsikas@ntnu.no; Tel.: +44-697-210732

**Abstract:** The automatic identification system (AIS), despite its importance in worldwide navigation at sea, does not provide any defence mechanisms against deliberate misuse, e.g., by sea pirates, terrorists, business adversaries, or smugglers. Previous work has proposed an international maritime identity-based cryptographic infrastructure (mIBC) as the foundation upon which the offer of advanced security capabilities for the conventional AIS can be built. The proposed secure AIS (SecAIS) does not require any modifications to the existing AIS infrastructure, which can still be used for normal operations. Security-enhanced AIS messages enjoying source authentication, encryption, and legitimate pseudo-anonymization can be handled on an as-needed basis. This paper reports on a proof-of-concept implementation of the SecAIS. Specifically, we report on the implementation of the SecAIS over an mIBC founded on the RFC6507 (ECCSI) and the RFC6508 (SAKKE) standards, and we discuss the results of performance tests with this implementation. The tests indicate that the SecAIS is a feasible solution that does not affect the conventional AIS infrastructure and has an affordable operational cost.

**Keywords:** automatic identification system (AIS); e-navigation; maritime security; identity-based encryption/cryptography

## 1. Introduction

Today's heavy marine traffic, caused by the booming maritime transportation, leisure, and fishing industry requires a large amount of trustworthy real-time navigational data to accurately reproduce the marine landscape around a vessel and to, therefore, ensure safe navigation. The radar, once the single-most valuable navigational aid, has a number of limitations: its accuracy is decreased in the presence of land obstacles, in bad weather conditions, when the target has a low radar-cross section (RCS), and when many targets are very close together.

To compensate for the above limitations, the maritime community uses supplementary systems for accurate vessel positioning and identification, such as the long-range identification and tracking (LRIT) system and the automatic identification system (AIS). Even though both LRIT and AIS provide positioning and identity information on the vessel, they have different aims and usage scopes. LRIT provides vessel data, through a rather complicated mechanism, only to contracting members, whilst the data broadcasted by the AIS are available to any receiver in range. Further, LRIT relies on satellite communication whilst AIS uses VHF as its main communication channel and offers the broadcasting of messages beyond the horizon as an add-on service via the satellite-AIS. LRIT transmits less information (e.g., date/time, vessel identity and position) than AIS, which broadcasts information such as voyage data, rate of turn, speed, and safety data [1]. Notwithstanding these advantages, the AIS is limited in scope and not mandatory for all vessels.

The International Maritime Organization (IMO), [2] in "Regulation 19" of the International Convention for the Safety of Life at Sea (SOLAS), Chapter V [3], announced the use of the AIS [4] as an additional navigational system. The International Telecommunications Union (ITU) [5] provided the technical characteristics of the AIS using time division multiple access in the very-high frequency (VHF) maritime mobile band. AIS is a shipborne device that transmits static data (e.g., Maritime Mobile Service Identity (MMSI), IMO number, call sign, ship name, type, vessel's dimensions), dynamic data (e.g., vessel's position (longitude, latitude), speed over ground (SOG), course over ground (COG), navigation status), voyage-related data (e.g., destination, estimated time of arrival, draught), and safety-related data [6]. The MMSI is a nine-digit number that uniquely identifies a vessel. The MMSI is assigned to all the radio communications of that vessel. The International Maritime Organization number (IMO number) is also a distinctive identifier for a vessel and is formed by the prefix "IMO" followed by seven digits. The main difference between the IMO number and the MMSI is that the former is the only persistent identifier for a vessel, from the start of its life to the end of it. On the contrary, the MMSI changes when a vessel changes flags and registration authorities.

While the accuracy of the data received from the radar is bound by the laws of physics, the accuracy of the data received by AIS devices depends on the trustworthiness of the transmitter. An adversary may transmit fake AIS messages representing fake ships, aids to navigation (AtoN), or search and rescue operations (SARs) [7]. Thus, the radar provides an honest, albeit possibly incomplete, representation of the surroundings of the ship, whilst AIS may give us a deliberately and maliciously distorted representation of it. Furthermore, AIS data are publicly available via specialized internet sites e.g., https://www.marinetraffic.com/, accessed on 11 June 2022, that collect forwarded AIS data from all over the world. This unrestricted dissemination of AIS data may come to aid sea pirates and terrorists, and in some cases, it may violate the privacy of passengers [8,9]. In fact, several incidents involving the AIS have been reported [10–12].

To thwart these threats, in [13], we proposed the deployment of an international maritime identity-based cryptographic (mIBC) infrastructure as the basis for enhancing AIS with extended modes of operation that provide on-demand anonymity, authentication, and encryption capabilities towards offering new security features such as message authentication, message encryption, and message pseudo-anonymization. In [13], we identified some scenarios where the usage of the proposed extended AIS modes of operation may support the safety of life at sea and the safety and efficiency of navigation. Specifically, the AIS data authentication and integrity mode of operation may shield the navigation from AIS spoofing attacks and fake AIS message attacks. The authenticated pseudo-anonymous AIS data mode of operation provides an alternative to turning off the AIS devices in situations where maintaining the anonymity of the vessel, and its cargo, is important (e.g., VIP yachts, vessels with sensitive cargo). Finally, in [13], we proposed the use of encrypted AIS data in insecure sea areas (e.g., areas where imminent danger for sea piracy exists) or in special situations (e.g., vessels carrying sensitive cargo, toxic waste, or weapons near coastlines or ports, where the threat of a terrorist attack is high) where the dissemination of AIS information would rather be controlled. For example, in insecure sea areas, access to the encrypted broadcasted AIS information will only be possible to vessels authenticated by the local authorities rather than to anyone (e.g., not authenticated sea pirates, terrorists, or internet sites).

In [14], we elaborated on the implementation aspects of that infrastructure and proposed its implementation by leveraging the work of Chen et al. [15] on Sakai–Kasahara schemes, on the work of Barreto et al. [16,17] on identity-based signatures, and on the IEEE 1363.3-2013 standard [18].

In this paper, we describe a proof-of-concept implementation of this secure AIS (SecAIS). Specifically, we describe a proof-of-concept implementation of the SecAIS over an mIBC founded on the RFC6507 [19] standard for Elliptic Curve-Based Certificateless Sig-

natures (ECCSI), and on the RFC6508 [20] for Sakai–Kasahara Key Encryption (SAKKE), and we demonstrate its workings.

The remainder of the paper is organized as follows: in Section 2, we discuss related work. Section 3 reviews our previous work in [13,14] to ensure the self-sustainability of the paper. In Section 4, we present the proof-of-concept implementation of the SecAIS. In Section 5, we demonstrate the workings of the implementation when sending and receiving SecAIS messages, and we show how to securely distribute the secret key of symmetric ciphers to multiple receivers in order to create encrypted AIS ad hoc networks (AISANETs) [13]. Section 6 summarizes our conclusions and outlines directions for future work.

## 2. Related Work

Several proposals for offering AIS-secure services have appeared in the literature. In [21], a security-enhanced AIS implementation that needs certificates from a cryptographic infrastructure was proposed. In [22], the IMO provided guidelines to promote the safe and effective use of shipborne automatic identification systems (AIS), in particular to inform the mariner about the operational use, limits, and potential uses of AIS. These guidelines do not improve the security of the AIS; they only show how to use the AIS as a communication channel to endow authentication capabilities to the ships. In [23], a global, x509-like Maritime PKI coordinated by the IMO and the National Maritime Authorities was proposed. Despite the advantages of the proposed solution, the global scale of the PKI and the use of resource-demanding certificates in the overloaded maritime wireless communication environment pose many implementation difficulties. The International Association of Marine Aids to Navigation and Lighthouse Authorities (IALA) presents, in its "e-Navigation Portal" [24], a number of proposals that may affect the future of AIS security. One of these proposals is to leverage the VHF Data Exchange System (VDES) [25] to address maritime security issues. However, the roadmap towards the universal adoption of the VDES is not yet clear; a debate in the maritime community about its deployment and operation cost is ongoing. In [26], the authors propose *SecureAIS*, a key agreement scheme that allows any pair of vessels in range of an AIS radio to agree on a shared session key of the desired length to be used for subsequent communications. This scheme allows for secure communication between two vessels only; thus, the broadcasting feature of AIS is not preserved. In a later paper [27], the same authors proposed Auth-AIS, a software-only solution that uses the TESLA authentication protocol and the Bloom filter tool to provide two modes of authentication, the "Deterministic Security Configuration' with a message overhead of 75% and the "Probabilistic Security Configuration" with a message overhead of 35%. According to the authors, the Auth-AIS, similar to the SecAIS proposed herein, is a software solution that offers authenticated AIS messages with similar security properties, cryptographic techniques, and small false rates. However, a message confidentiality service is not offered. The TESLA protocol has also been leveraged to propose an authentication protocol that enhances the security of the AIS by providing it with a message-integrity and broadcast-authentication feature [28]. Kessler [29] has proposed a name-protected-AIS (pAIS), a concept similar to the authentication mode (mode 2) of the proposed SecAIS. However, the use of a 256-RSA key may lead to a private key compromise. Further, the use of RSA carries over the problems of certificate-based public key infrastructures. The use of public key cryptography (PKC) for authenticating AIS messages is also proposed in [30]. However, a separate VHF Data Exchange System (VDES) side channel is required to carry PKC digital signatures. Several backwards-compatible signature schemes for AIS messages are reviewed and compared concerning their applicability in [31].

Identity-based encryption (IBE) schemes are certificateless public key cryptosystems that were originally proposed by Shamir in 1985 [32]. Their advantage is that their public and private keys are derived from the identifiers of the participating entities [33–36] (e.g., their MMSI numbers, as in [13,14]). IBE schemes were also proposed in [37–39] to address the security issues of the automatic dependent surveillance–broadcast (ADS-B) system,

an aviation system with functionality similar to that of the AIS. However, these proposed solutions cannot be transferred to the maritime environment or to the case of the AIS, as they have been designed for the specific ADS-B environment.

An example of a non-commercial IBE scheme is proposed in RFC6508, which defines an identity-based encryption (IBE) implementation based on elliptic curves. This IBE scheme is optimized to be used for both the Sakai–Kasahara key encryption (SAKKE) algorithm, described in RFC6508, and the certificateless signatures for identity-based encryption (ECCSI) signature scheme, described in RFC6507. RFC6508 uses IBE to securely disseminate a "shared secret" (e.g., the key of a symmetric cipher) to a receiver, whilst RFC6507 provides authentication via certificateless signatures that are based on the Elliptic Curve Digital Signature Algorithm (ECDSA). In particular, RFC6508 (SAKKE) presents a variant of the Sakai–Kasahara key encapsulation mechanism (SK-KEM), optimized to support multi-party communications that have been adopted by the IEEE 1363.3 2013 standard for identity-based cryptography. The RFC6507 standard presents a certificateless variant of the Elliptic Curve Digital Signature Algorithm (ECDSA) that is designed to be used with identity-based encryption; it is compatible with the IBE proposed in RFC6508, and is optimized to have low bandwidth and low computational requirements.

In 2016 the National Cyber Security Centre of the UK proposed the MIKEY-SAKKE protocol to provide secure, cross-platform multimedia communications (e.g., voice over IP) for government-related agencies. Their proposal was presented in RFC6509 [40]; it uses IBE/IBC, the Sakai–Kasahara key encryption (SAKKE) algorithm and a variant of the ECDSA adapted to be used with the Sakai–Kasahara protocol.

### 3. Maritime Identity-Based Cryptographic Infrastructure (mIBC)

In [13], we proposed the deployment of a maritime identity-based cryptographic infrastructure (mIBC) that enhances the AIS with on-demand anonymity, authentication, and encryption capabilities. In [13,14], the term "mIBC-AIS" was used instead of the term "SecAIS". To maintain consistency, all "mIBC-AIS-" terms in [13,14] have been replaced by "SecAIS-" terms herein. In [14], we proposed the implementation of the mIBC infrastructure following the IEEE 1363.3-2013 standard for identity-based cryptography, and we defined five distinct usage modes; each of these can be used according to the needs of the SecAIS user at any time. These modes are:

1. The **Typical-SecAIS (mode 1)** is the conventional AIS, for routine use;
2. The **Authenticated-SecAIS (mode 2)** offers source authentication via cryptographically signed AIS messages. An AIS device signs the transmitted AIS data with its mIBC private (secret) key, and the receivers authenticate the signed AIS messages by using only the MMSI of the transmitter vessel;
3. The **Anonymous-SecAIS (mode 3)** offers legitimate anonymous AIS-transmitted messages via "pseudo-MMSIs" that are cryptographically signed by an official mIBC agency. From a cryptographic point of view, the Anonymous-SecAIS (mode 3) is identical to the Authenticated-SecAIS (mode 2), although it uses a pseudo-MMSI instead of the real MMSI of the vessel;
4. The **SK-IBE-SecAIS (mode 4)** allows for the secure transmission of small encrypted AIS messages to a specific entity via an appropriate encryption scheme, such as the Sakai–Kasahara identity-based encryption scheme. It is proposed to be used, mainly, for the secure sharing (or distribution, if there is more than one recipient) of the symmetric keys in the AES-SecAIS (mode 5);
5. The **AES-SecAIS (mode 5)** allows for the transmission of encrypted AIS messages to a group of participants (e.g., trustworthy vessels in an insecure area) via symmetric cryptography (e.g., by means of the advanced encryption standard (AES) [41]). The advantage of our solution is that the keys for the encryption can be generated and disseminated ad hoc, upon request, unlike the current commercially available solutions, e.g., [42,43], where the keys are pre-configured in predetermined users.

This approach allows for the ad hoc creation of encrypted AISANETs, described in Section 5.

To enhance the conventional AIS devices with SecAIS capabilities, in [14], we proposed the use of a special add-on application named "SecAIS-App". This is located between the conventional AIS devices and the AIS transmitter, and it implements the security modes of the SecAIS by acting as the cryptographic interface between current AIS devices. It works as follows:

1.  The SecAIS-App intercepts the original generated AIS message (e.g., an AIS class A ship static and voyage-related data ID5 [5]);
2.  The SecAIS-App performs the appropriate cryptographic actions (e.g., signing, encrypting, etc.);
3.  The SecAIS-App creates conventional AIS binary broadcast messages of types ID8 or ID 6 [5], and it encapsulates the SecAIS data into their "application data payload" section. The SecAIS-App forwards the newly created, conventional AIS binary broadcast messages to the AIS transmitter.

Similar actions, in reverse order, are performed by the SecAIS-App of the receiver. Therefore, the SecAIS is entirely transparent to current AIS devices. Figure 1 depicts the use of the SecAIS-App and its interaction with the AIS devices when transmitting/receiving an authenticated SecAIS message (mode 2).

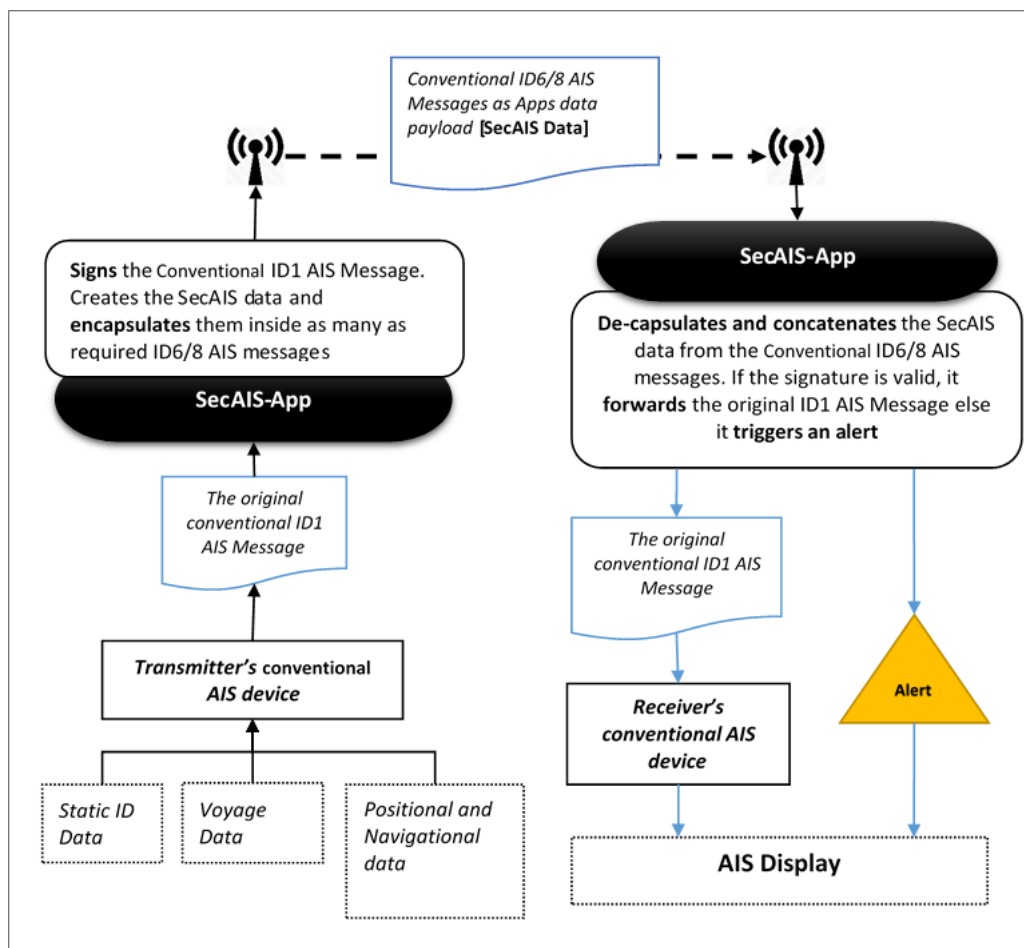

**Figure 1.** Authenticated SecAIS (mode 2).

## 4. Proof-of-Concept Implementation

The proof-of-concept implementation described in this section is designed to operate in an environment closely resembling a real maritime environment. Key aspects of the implementation are:

- The mIBC infrastructure (see Section 4.1) is implemented using the IBC model and the cryptographic values specified in the "Appendix A: Test Data" of RFC6507 and RFC6508;
- We use conventional ID6/8 AIS messages as the carriers of the mIBC cryptographic parameters (see Section 4.2);
- We simulate conventional AIS devices by using a third-party online AIS VDM/VDO decoder (see Section 4.3);
- The SecAIS-App is coded in Java using third-party code (see Section 4.4).

### 4.1. The mIBC Infrastructure

The mIBC infrastructure was described in detail in [13,14]. A similar identity-based cryptographic (IBC) model is presented in RFC6507 and RFC6508. In particular, in "Appendix A: Test Data", these RFCs provide an IBC model with specific cryptographic parameters for testing purposes. This IBC model, along with its cryptographic values, is used herein to implement the mIBC-AIS infrastructure. By doing so:

- We are able to check the validity of all cryptographic computations in this work by simply comparing our results to those in "Appendix A: Test Data" of RFC6507 and RFC6508;
- We implement the mIBC in an over-demanding cryptographic environment. This is because both RFC6507 and RFC6508 adopt a security level higher than what is necessary for the mIBC. Therefore, by using the cryptographic values of RFC6507 and RFC6508, we demonstrate the workings of our solution in the worst-case scenario, with sizes of the transmitted cryptographic data larger than the ones required in the real world. For example, in the proposed mIBC, the ID of each vessel would be its nine-digit MMSI; in the proof-of-concept implementation, we use the ID in "Appendix A: Test Data" of RFC6507 and RFC6508, namely, the (much longer) ID "2011-02\0tel: +447700900123\0".

The exact values of each cryptographic parameter that are used in this paper can be found in "Appendix A: Test Data" of RFC6507 and RFC6508. To maintain consistency with the nomenclature used in [13,14], Table 1 depicts the cryptographic components of the mIBC of [13,14] and the corresponding cryptographic parameters of RFC6507 and RFC6508.

**Table 1.** Cryptographic components.

| Parameter | mIBC | PoC Implementation |
|---|---|---|
| Private Key Generator | IMO-mIBC-PKG | PKG |
| mIBC Public Parameters | IMO-mIBC-PKG-PP | Public Parameters of the RFCs |
| Public key | MMSI | ID = "2011-02\0tel: +447700900123\0" |
| Private key | IMO-mIBC-PKG-SKMMSI | Secret-key (SSK) |

### 4.2. Using ID6 and ID8 AIS Messages as Carriers of the SecAIS Data

The cryptographic data of the SecAIS are transmitted over the conventional AIS infrastructure, by means of either ID6 or ID8 AIS messages. ID6 AIS messages are designed for addressed communication (i.e., to a specific MMSI), and ID8 AIS messages are designed for broadcast communication. Both offer an isolated data payload subsection of 920/952 bits, designed to carry non-AIS-related data for registered third-party applications. This approach has been time-tested by many applications and has the advantage of being completely transparent to the underlying AIS infrastructure. However, because the size of the data carried by each ID6/8 AIS message is limited, more than one ID6/8 AIS message

is required to transmit SecAIS data. In this implementation, we use ID8 AIS messages for transmitting both Authenticated-SecAIS (mode 2) and SK-IBE-SecAIS (mode 4) data. It must be noted that, depending on the SecAIS usage mode, the transmission of SecAIS cryptographic data requires the following number of ID6/8 AIS message(s):

- The transmission of Authenticated-SecAIS (mode 2) data requires two ID8 AIS messages. The first is an initial conventional ID8 AIS message to send the public key (PVT) that is derived from the MMSI of the vessel. The second is the conventional ID8 AIS message that transfers the signed version of the original AIS message;
- The transmission of SK-IBE-SecAIS (mode 4) data requires three ID8 AIS messages per receiver of the secret key for the symmetric cipher.

### 4.3. The AIS Devices

To demonstrate that the proposed SecAIS implementation works transparently over the conventional AIS infrastructure without affecting its operation, an online AIS VDM/VDO decoder (https://www.maritec.co.za/tools/aisvdmvdodecoding/, accessed on 11 June 2022)is used to simulate conventional AIS devices. The specific decoder was chosen because, unlike others, it permits the decoding of AIS ID6/8 AIS messages that carry non-registered application data.

### 4.4. The SecAIS-App

The SecAIS-App is the application that enhances conventional AIS devices with SecAIS capabilities. For the purposes of this paper, the code implementing the SecAIS-App supports only the Authenticated-SecAIS (mode 2) and the SK-IBE-SecAIS (mode 4). It is written in Java and it permits the integration of arbitrary open-source code and cryptographic libraries. We use the open-source Java cryptographic libraries provided by the "Legion of the Bouncy Castle" (https://www.bouncycastle.org/java.html, accessed on 11 June 2022), which provides robust and well-preserved Java cryptographic libraries with an MIT-type license. Some parts of the SecAIS-App implementation are copied from or are based on the jim-b/ECCSI-SAKKE project on GitHub (https://github.com/jim-b/ECCSI-SAKKE, accesed on 11 June 2022), which is part of an open-source implementation of ECCSI-SAKKE (RFC6507-RFC6508), MIKEY-SAKKE, and the MIKEY-SAKKE key management server (KMS) (RFC6509). In a SecAIS production environment, each AIS device will be equipped with a SecAIS-App that will act as both a SecAIS transmitter and receiver.

## 5. How the SecAIS Works

### 5.1. SecAIS Source Authentication

In Authenticated-SecAIS (mode 2), each AIS device will be able to sign the broadcasted AIS messages with its mIBC private key (SSK); the receivers will be able to authenticate the signed AIS data by using the MMSI of the broadcasting vessel. In brief, the above scenario works as follows (see Figure 2):

1. The vessel that broadcasts the AIS message (hereafter "the Transmitter"), uses the SecAIS-App to sign its positional ID1 AIS message before it is transmitted;
2. Any AIS receiver (hereafter "the Receiver") uses its SecAIS-App to validate the signature and, thus, the authenticity of the received positional ID1 AIS message.

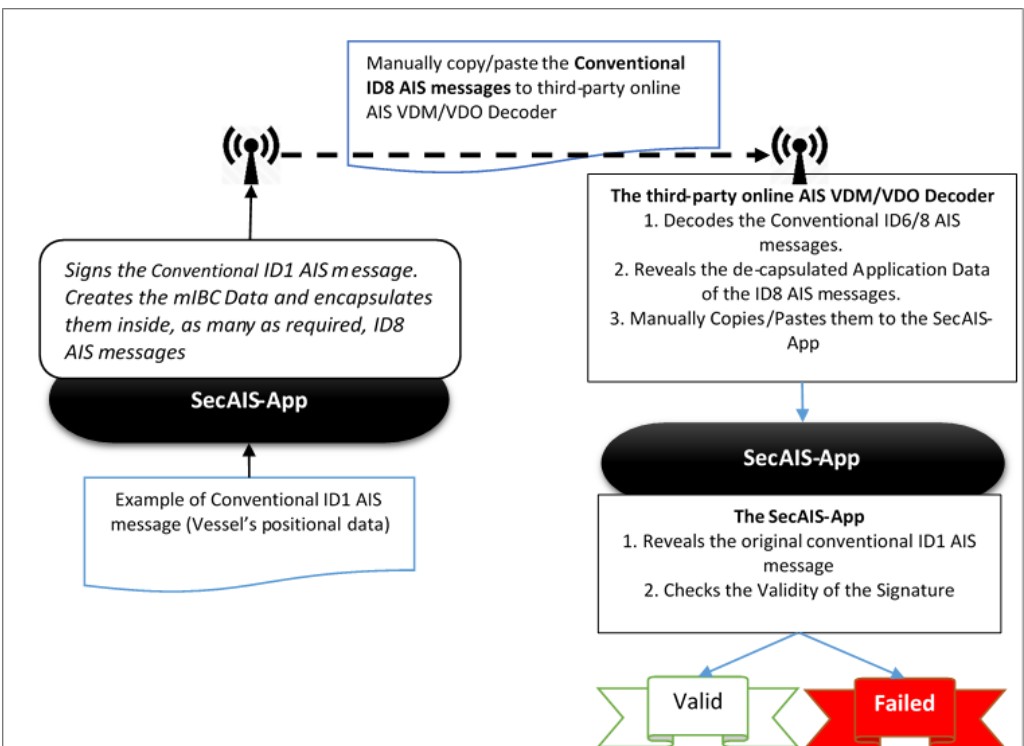

**Figure 2.** Simulated operation of the Authenticated SecAIS (mode 2) using ECSSI (RFC6507).

### 5.1.1. Creating the SecAIS Message

Assume that the vessel with MMSI = "2011-02\0tel: +447700900123\0" uses the SecAIS to cryptographically sign and transmit the original conventional ID1 AIS message depicted in Figure 3. Note that, in a SecAIS production infrastructure, the MMSI of the Transmitter should be the same as the MMSI of the original ID1 AIS message ("265547250" in the depicted AIS message). This is necessary because the source authentication is validated against the **claimed**ID/MMSI of the ID8 AIS message that encapsulates the signed AIS message (i.e., the original ID1 AIS message).

The process is as follows: the AIS device of the Transmitter generates the original AIS message for broadcasting. Before transmission, the SecAIS-App intercepts the message and creates the corresponding authentication signature. In this implementation, the original ID1-AIS message (i.e., "!AIVDM,1,1,,A,13u?etPv2;0n:dDPw UM1U1Cb069D,0*24" in the red rectangle in Figure 3) is copied to the SecAIS-App (see Figure 4, part A)]. The SecAIS-App automatically prepends a timestamp and appends the authentication signature. The final message to be signed is the concatenation of the timestamp, the original AIS message, and the signature; see Figure 4, part C. The addition of the signed timestamp is a counter-measure to a potential replay attack, where an adversary may attempt to impersonate a legitimat vessel by using a recorded signed SecAIS message from the victim. Therefore, a SecAIS message needs to have both a valid signature and a valid timestamp to be valid.

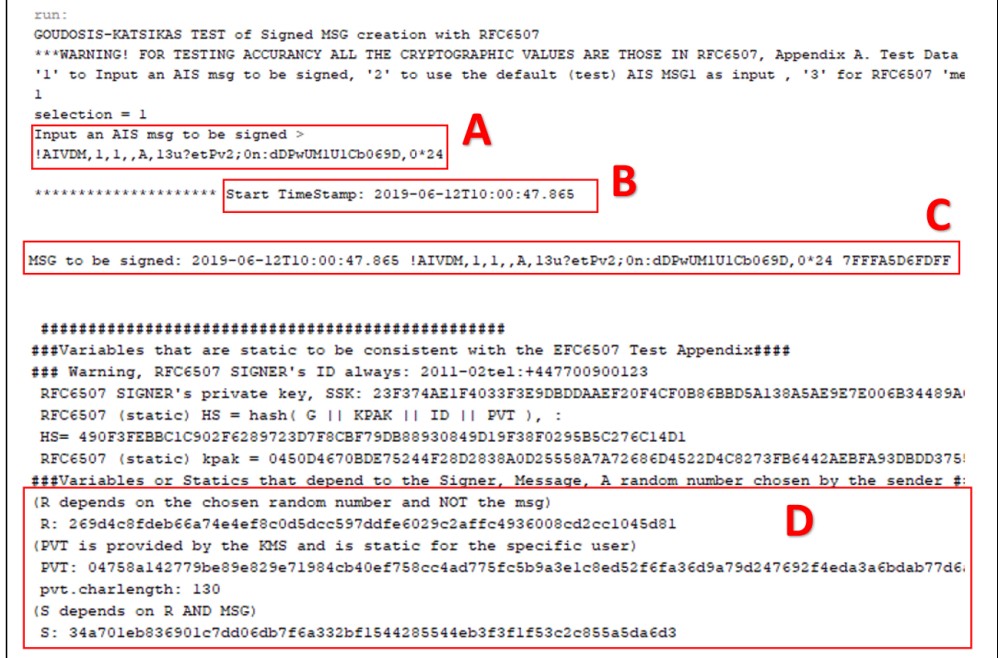

**Figure 3.** Original conventional ID1 AIS message.

**Figure 4.** Original ID1 AIS message (**A**); timestamp (**B**); message to be signed (**C**); signature (**D**).

### 5.1.2. Encapsulating the Signature in Conventional ID8 AIS Messages

Next, the SecAIS-App of the Transmitter encapsulates the signature and the original AIS message into two conventional ID8 AIS messages. This is carried out in three steps, as follows:

1.  Encapsulate the PVT part of the signature inside the first conventional ID8 AIS message, as shown in Figure 5;
2.  Encapsulate the original signed message, the timestamp, and the parameters R and S of the signature inside the second conventional ID8 AIS message, as shown in Figure 6;
3.  Perform a validation check of the correctness of the signature before broadcasting the two ID8 AIS messages, as shown in Figure 7.

```
^^^^^^^^^^^^^^^^^^^^^^^^^^^^^^^^^^^^^^^^^^^^^^^^^^^^^^^^^^^^^^^^^^^^^^^^^^^^^^^^^^^^
(DMSG8_Msg_R_S_PVT_Creation_v5)
total_AIS_MSG8_DataPayload.length() 134

****** Copy/Paste the Msg8-NMEA_AIS below that conatins:

>>>> This MSG8 contains: The static 'PVT' of the Signer <<<< *****

!AIVDM,3,1,,B,81mg=5@0H45U?L==uN0LM<euv@QN>AN<fAMtNN=0he<1AeuN0hu0A=uuAPu@f@,4*5A
!AIVDM,3,2,,B,81mg=5@0H0LiL@v1A=LQeQPLuQ>@MvA<e=ufLQe1A0LhMPQ0@eui=P@Me=u0M0,4*0A
!AIVDM,3,3,,B,81mg=5@0H=e0AM>Lu=edhuLee@PMt<N0PL>LAev@,4*0B
```

**Figure 5.** ID8 AIS message that encapsulates the PVT part of the signature.

```
^^^^^^^^^^^^^^^^^^^^^^^^^^^^^^^^^^^^^^^^^^^^^^^^^^^^^^^^^^^^^^^^^^^^^^^^^^^^^^^^^
(DMSG8_Msg_R_S_PVT_Creation_v5)
total_AIS_MSG8_DataPayload.length() 218

****** Copy/Paste the Msg8-NMEA_AIS below that conatins:
This MSG8 contains: The original AIS Msg1 + From the signing process the the 'R' and 'S'

!AIVDM,5,1,,B,81mg=5@0H<d<NKL=cLLU<L>d<>e=sf=eH8@BEQ3K<K<K;0K<LmOiE45dft3fQ0,4*6E
!AIVDM,5,2,,B,81mg=5@0H145mCLEL@hd=fA;<:de8=iQQPMA=QQ1Q`4gLefA=0v1Q1@eePMu1@,4*07
!AIVDM,5,3,,B,81mg=5@0H=1Af0t1=A0huNMi11QMd<f@tPAQPu>Lud<>0i<PhtL==A><H4wLu0,4*47
!AIVDM,5,4,,B,81mg=5@0H0Mt<A@f<ufL<@uil<=Q0eiePLttPQdMM=<f=MM=1@dididAeLhtPh,4*3D
!AIVDM,5,5,,B,81mg=5@0H>=M@MA0MQ<h,4*68
```

**Figure 6.** ID8 AIS message that encapsulates the original signed message, the added timestamp, and the variable cryptographic parameters of the signature.

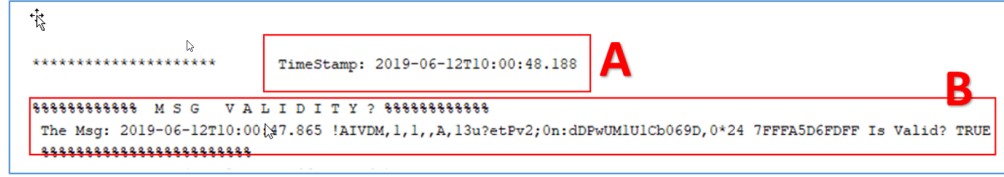

**Figure 7.** Timestamp (**A**) and signature validity check (**B**).

The time taken to complete the above process is negligible: in our experiments, it took less than a second to complete on a PC with the following characteristics: processor—Intel Xeon ®; CPU—E5-1620 Vv2 @ 3.70 GHz; RAM—16 GB; OS—64-bit Windows 10 Pro. Specifically, the process started at 2019-06-12T10:00:47.865 (Figure 4 (part B)) and ended at 10:00:48.188 (Figure 7).

### 5.1.3. Retrieving the Original AIS Message

All the conventional AIS devices that receive the two broadcasted ID8 AIS messages are able to decode them and retrieve the SecAIS application data. The sequence of actions of the conventional AIS device of the Receiver for doing so are as follows:

1.  Step 1: receives the encoded ID8 AIS message containing the PVT of the Transmitter;
2.  Step 2: decodes the encoded ID8 AIS message containing the PVT of the Transmitter;
3.  Step 3: receives the encoded ID8 AIS message containing the original signed message, the timestamp, and the variable cryptographic parameters of the signature (R, S);
4.  Step 4: decodes the received ID8 AIS message.

Screenshots depicting the various stages of the process are shown in Figures 8–10. The conventional AIS device of the Receiver is again simulated by the online AIS VDM/VDO decoder as it was on 6 October 2019; the browser used was Firefox© Quantum version 67.0.1 (64 bit).

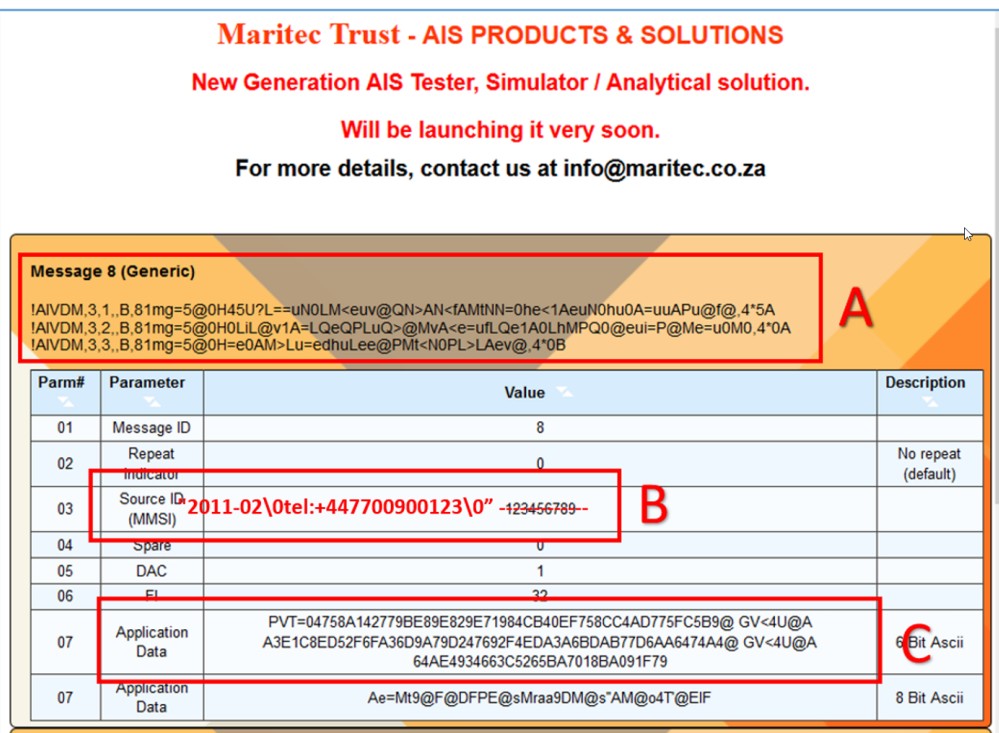

**Figure 8.** ID8 AIS message (**A**); PVT of the Transmitter (**C**); source MMSI (**B**).

## AIS VDM/VDO Decoder

### ENTER ONE MESSAGE PER LINE. MULTI SENTENCE MESSAGES HAVE TO BE ENTERED IN DIRECT SEQUENCE (viz. MSG 5, Msg 1-of-2, 2-of-2 )

*Checksum (xor) failure is indicated where applicable. Using this decoder, you have accepted the Terms & Conditions.*

```
!AIVDM,5,1,,B,81mg=5@0H<d<NKL=cLLE<LNdMNdM;d=tH8@BEQ3K<K<K;
0K<LmOiE45dft3fQ0,4*25
!AIVDM,5,2,,B,81mg=5@0H145mCLEL@hd=fA;
<:de8=iQQPMA=QQ1Q`4gLefA=0v1Q1@eePMu1@,4*07
!AIVDM,5,3,,B,81mg=5@0H=1Af0t1=A0huNMi11QMd<f@tPAQPu>Lud<>0i<PhtL==A>
<H4w@A@,4*0F
!AIVDM,5,4,,B,81mg=5@0H1Ld=<@L0d=QMQfAAQ@L1@Pt=<t=uL@fMLMteLd-
ddMe<d0LhuN0i<@,4*15
!AIVDM,5,5,,B,81mg=5@0H1=@MdeN<PiP,4*09
```

| Decode | Clear Textbox |
|--------|---------------|

**Figure 9.** ID8 AIS Message that encapsulates the original signed message, the added timestamp, and the variable cryptographic parameters of the signature.

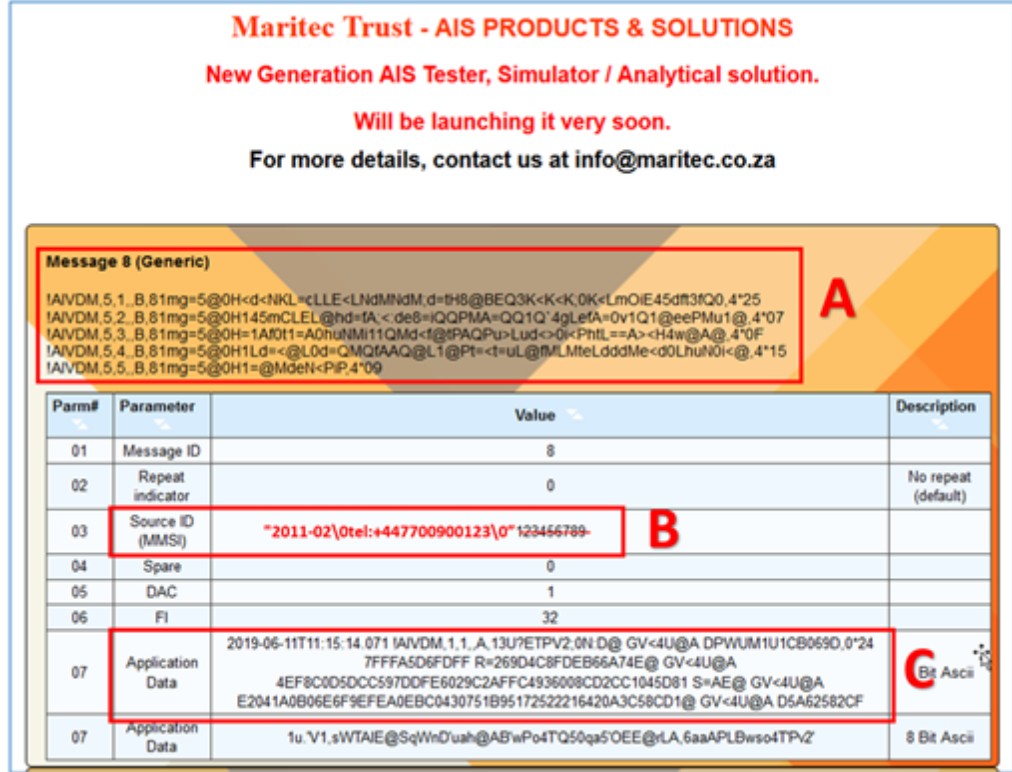

**Figure 10.** Original signed message (**A**); timestamp (**B**); variable cryptographic parameters of the signature (**C**).

### 5.1.4. Validating the Signature

Following the retrieval of the signed SecAIS message, the SecAIS-App checks the validity of the authentication signature against the signed AIS message and the signed timestamp. Figure 11 shows this process in an experiment where the result of the signature validation is "TRUE". Note that, in a real implementation, the ID8 AIS message depicted in these figures will fail to pass the signature validation procedure, because the source MMSI = "1234567879" differs from the MMSI = "2011-02\0tel: +447700900123\0", which is bound to the cryptographic values R, S, and PVT. This is why, for the purposes of the experiment, we manually input the correct MMSI = "2011-02\0tel: +447700900123\0" to the SecAIS-App. The result of a test where a malicious AIS transmitter uses a spoofed MMSI (i.e., MMSI = "THIS IS A SPOOFED ID") instead of the real one (i.e., MMSI = "2011-02\0tel: +447700900123\0" in this implementation) are shown in Figure 12; the result of the validation is "FALSE", as it should be.

The time taken for the signature validation process to complete is negligible: on a PC with CPU: Intel Xeon ® E5-1620 Vv2 @ 3.70 GHz, RAM: 16 GB, and OS: 64-bit Windows 10 Pro, the signature validation process took less than a second to complete. Specifically, the valid signature test started at 11:38:16.889 and ended at 11:38:17.038 (Figure 11, Parts C), and the invalid signature test started at 11:40:10.422 and ended at 11:40:10.576 (Figure 12, part D).

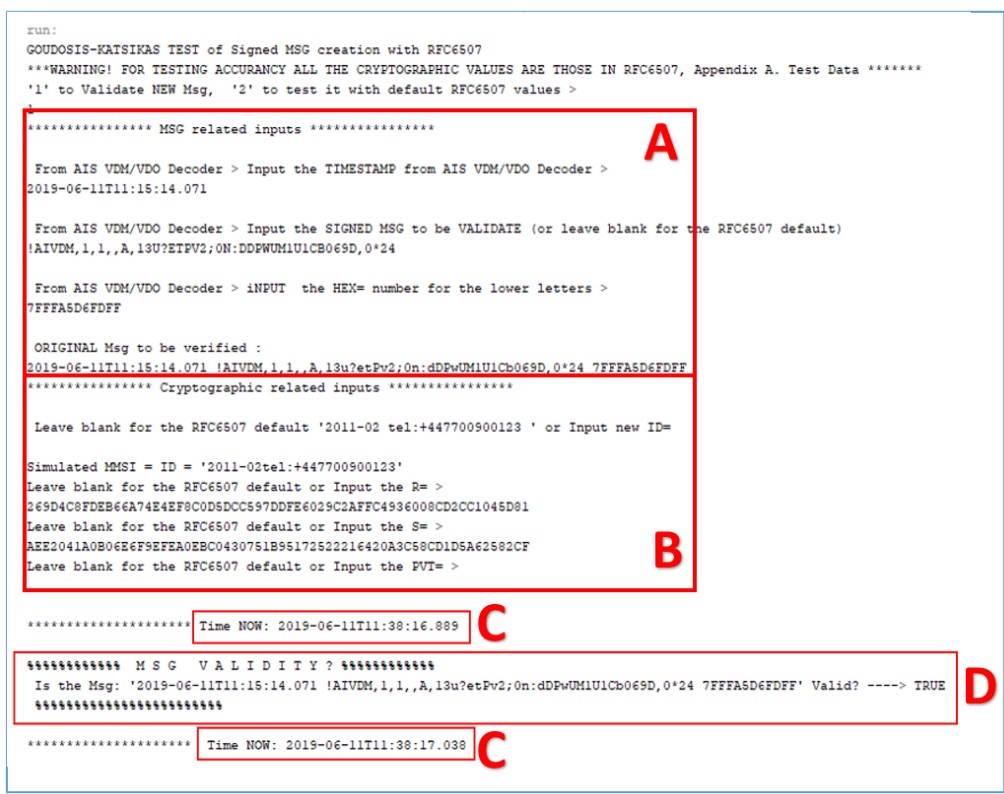

**Figure 11.** Authentic and valid SecAIS message: Message-related inputs (**A**); Cryptographic-related inputs (**B**); Start and end timestamps of validity check (**C**); Result of validity check (**D**).

```
GOUDOSIS-KATSIKAS TEST of Signed MSG creation with RFC6507
***WARNING! FOR TESTING ACCURANCY ALL THE CRYPTOGRAPHIC VALUES ARE THOSE IN RFC6507, Appendix A. Test Data *******
'1' to Validate NEW Msg,  '2' to test it with default RFC6507 values >
1
*************** MSG related inputs ***************

 From AIS VDM/VDO Decoder > Input the TIMESTAMP from AIS VDM/VDO Decoder >
2019-06-11T11:15:14.071

 From AIS VDM/VDO Decoder > Input the SIGNED MSG to be VALIDATE (or leave blank for the RFC6507 default)
!AIVDM,1,1,,A,13U?ETPV2;0N:DDPWUM1U1CB069D,0*24

 From AIS VDM/VDO Decoder > iNPUT  the HEX= number for the lower letters >
7FFFA5D6FDFF

 ORIGINAL Msg to be verified :
2019-06-11T11:15:14.071 !AIVDM,1,1,,A,13u?etPv2;0n:dDPwUM1U1Cb069D,0*24 7FFFA5D6FDFF
*************** Cryptographic related inputs ***************

 Leave blank for the RFC6507 default '2011-02 tel:+447700900123 ' or Input new ID=
 'THIS IA A SPOOFED ID'
Leave blank for the RFC6507 default or Input the R= >
269D4C8FDEB66A74E4EF8C0D5DCC597DDFE6029C2AFFC4936008CD2CC1045D81
Leave blank for the RFC6507 default or Input the S= >
AEE2041A0B06E6F9&FEA0EBC0430751B95172522216420A3C58CD1D5A62582CF
Leave blank for the RFC6507 default or Input the PVT= >

********************* Time NOW: 2019-06-11T11:48:10.422

$$$$$$$$$$$$$ M S G   V A L I D I T Y ? $$$$$$$$$$$$$
 Is the Msg: '2019-06-11T11:15:14.071 !AIVDM,1,1,,A,13u?etPv2;0n:dDPwUM1U1Cb069D,0*24 7FFFA5D6FDFF' Valid? ----> FALSE
 $$$$$$$$$$$$$$$$$$$$$$$$$$

********************* Time NOW: 2019-06-11T11:48:10.576
```

**C**

**D**

**Figure 12.** Invalid SecAIS message: Input of the SpoofedID (**C**); "FALSE" outcome (**D**).

*5.2. Sharing Secrets in SecAIS*

This section describes the workings of the SK-IBE-SecAIS (mode 4), which enables conventional AIS devices to send a secret (e.g., a key for a symmetric cipher) to a specific Receiver. In this implementation, the shared secret (i.e., the key of a cipher) to be transmitted is the one in "Appendix A: Test Data" of RFC6508. Since the process is similar to that described in Section 5.1, we only highlight the main stages:

1. The SK-IBE-SecAIS (mode 4) data are separated into three distinct strings, namely H, Rbx, and Rby; details on these can be found in RFC6508. Since the SecAIS data cannot fit into a single ID8/6 AIS message, the SecAIS-App creates three ID8/6 AIS messages, each containing the H, Rbx, and Rby strings, respectively; see Figure 13;
2. The conventional AIS device of the Transmitter transmits these messages (Figure 13, parts H, Rbx, RBy), which are addressed to the conventional AIS device of the Receiver. Figure 14 depicts the decoded ID8 AIS message that contains the H string. Similarly, the Receiver decodes and reformats the Rbx and Rby strings;
3. The SecAIS-App of the Receiver uses the decoded and reformatted H, Rbx, and Rby strings to recreate the secret (e.g., the key for a symmetric cipher).

The time taken for the signature validation process to complete is negligible: on a PC with CPU: Intel Xeon ® E5-1620 Vv2 @ 3.70 GHz, RAM: 16 GB, and OS: 64-bit Windows 10 Pro, the signature validation process took less than a second to complete. Specifically, the signature test started at 09:24:53:610 and ended at 09:24:53:636 (Figure 13, parts D and C).

```
********************* Start TimeStamp: 2019-06-28T09:24:53.610     D

^^^^^^^^^^^^^^^^^^^^^^^^^^^^^^^^^^^^^^^^^^^^^^^^^^^^^^^^^^^^^^^^^^^^^^^^^^^^^^^^^
 (DMSG8_Msg_R_S_PVT_Creation_v5)
 total_AIS_MSG8_DataPayload.length() 32

****** Copy/Paste the Msg8-NMEA_AIS below that conatins:

 >>>> This MSG8 contains: The H *****      H

!AIVDM,1,1,,B,8lmg=5@0H>>AL0Pued@@LANLMdvlMP@hv=lM>MeL=h,4*67

 (MSG8_Msg_R_S_PVT_Creation_v5) NMEA_AIS_Msg8_final_result.length(): 61 characters

_______________________________________________

 ^^^^^^^^^^^^^^^^^^^^^^^^^^^^^^^^^^^^^^^^^^^^^^^^^^^^^^^^^^^^^^^^^^^^^^^^^^^^^^^^
 (DMSG8_Msg_R_S_PVT_Creation_v5)
 total_AIS_MSG8_DataPayload.length() 256

****** Copy/Paste the Msg8-NMEA_AIS below that conatins:

 >>>> This MSG8 contains: The Rbx *****     Rbx

!AIVDM,5,1,,B,8lmg=5@0H==lN0A==0@f=NLPMPM@Lil0hM@if>MPutN<=<ud=PL<A=eLllAduh,4*0F
!AIVDM,5,2,,B,8lmg=5@0H0L<Aduhtdf0tttQPttMtuMlLPteul=l0Af0A<<<L=M0uuPiMMvMt@,4*4B
!AIVDM,5,3,,B,8lmg=5@0H0uQe=>=Q=Mtdt=<ted@uL=Q@Qe@QM<vleLilL=ld=ehuuuQLll<hP,4*25
!AIVDM,5,4,,B,8lmg=5@0H=t@MdfL<Lttf<ute@MLtQdd@AdM=LLeQ0tA=uui@htehQML><u@Q0,4*14
!AIVDM,5,5,,B,8lmg=5@0H<f<>N0f0Mti>Af<<A>>LuvLhM<AQe@u>@f=iMvAdPQMl=MPi@,4*16

 (MSG8_Msg_R_S_PVT_Creation_v5) NMEA_AIS_Msg8_final_result.length(): 401 characters

_______________________________________________

 ^^^^^^^^^^^^^^^^^^^^^^^^^^^^^^^^^^^^^^^^^^^^^^^^^^^^^^^^^^^^^^^^^^^^^^^^^^^^^^^^
 (DMSG8_Msg_R_S_PVT_Creation_v5)
 total_AIS_MSG8_DataPayload.length() 256

****** Copy/Paste the Msg8-NMEA_AIS below that conatins:

 >>>> This MSG8 contains: The Rby *****     Rby

!AIVDM,5,1,,B,8lmg=5@0H=MMiLLu0A>=@PdA=0f@iMlf0QM0d>0LLPP@QeM@dA=QdA=hMdv<<@,4*01
!AIVDM,5,2,,B,8lmg=5@0H>A@LflLM@@d@vAeuduuAQl<LdLl=le<uL@f@L<>M>=PeidiA==Pv@,4*63
!AIVDM,5,3,,B,8lmg=5@0H=eAlA<Q><l0AlMldv0uetdA=LPthA<L<hLLl@Q<fMNLe>0elAd<=P,4*5A
!AIVDM,5,4,,B,8lmg=5@0H<uPQd>Mu=>lMdL=hvAAlANAPet=><dildNNAf<tPvL=@AM=Af0LP,4*68
!AIVDM,5,5,,B,8lmg=5@0H==hL=tQ>lAetfA@@PuALeuMu0d=utv@du0QMu0MLtQeu=hf=P,4*1A

 (MSG8_Msg_R_S_PVT_Creation_v5) NMEA_AIS_Msg8_final_result.length(): 401 characters

_______________________________________________

********************* End TimeStamp: 2019-06-28T09:24:53.636      C
```

**Figure 13.** Creation of the three conventional ID8 AIS messages containing the H, Rx, and Ry cryptographic values.

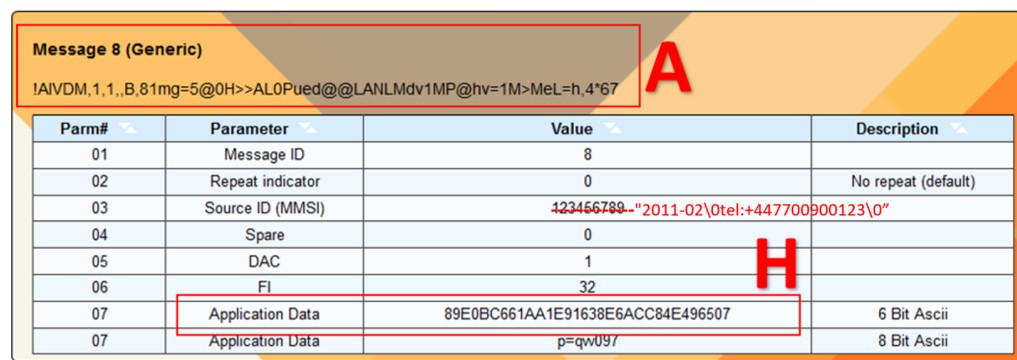

**Figure 14.** Cryptographic value H (**H**) on the application data section of the conventional ID8 AIS message (**A**).

## 6. Conclusions

In this work, a proof-of-concept implementation of the SecAIS in an environment simulating the conventional AIS infrastructure was presented. The cryptographic infrastructure is based on RFC6507 and RFC6508; a third-party online AIS VDM/VDO decoder was used as the conventional AIS device. The main conclusion of this work is that the SecAIS is a viable option for offering authenticated AIS transmissions and ad hoc encrypted AIS transmissions of small secrets (e.g., keys for symmetric ciphers) to the maritime community. Furthermore, this is achieved without modifying the conventional AIS infrastructure or affecting its everyday use at a negligible operational cost.

Our future work will focus on minimizing the size of the SecAIS data. Since the size of these data depends on the cryptographic parameters used, an obvious solution is to lower the level of security offered by the SecAIS. However, an optimal trade-off between the level of security that the SecAIS offers and its operational cost should be sought.

**Author Contributions:** Conceptualization, A.G. and S.K.; methodology, A.G.; software, A.G.; validation, A.G; resources, A.G. and S.K.; writing—original draft preparation, A.G.; writing—review and editing, S.K.; supervision, S.K.; project administration, S.K. All authors have read and agreed to the published version of the manuscript.

**Funding:** This research was funded, in part, by the Research Council of Norway, under Project number 310105 "Norwegian Center for Cybersecurity in Critical Sectors (NORCICS)".

**Institutional Review Board Statement:** Not applicable

**Informed Consent Statement:** Not applicable

**Data Availability Statement:** Not applicable

**Conflicts of Interest:** The authors declare no conflict of interest.

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
