# Peer review of "Secure Automatic Identification System (SecAIS): Proof-of-Concept Implementation"

_jmse, doi:10.3390/jmse10060805_

Round 1
Reviewer 1 Report
1. According to the IMO Resolution A.1106(29) AIS is intended to enhance: safety of life at sea and the safety and efficiency of navigation. It shall support ship radar and radar tracking system by assisting in: identification of targets and their navigational status, presentation of targets heading, immediate identification of manoeuvres and more accurate presentation of the targets vectors. As a tool used for this purpose, it can be installed on any convention (AIS class A) or non-convention ship (e.g. AIS class B and AIS receiving device) and it will be difficult to prohibit its installation on any type of vessel. The basic data transmitted by the ship's AIS (ship identification, position, course, speed, navigational status) should be understandable for each AIS, regardless of its class, installed on another ship. Knowing these legal conditions, the authors should, in my opinion, write how the security features proposed in the article will prevent a pirate or terrorist attack carried out from another ship equipped with AIS.
2. The SOLAS convention was adopted only by the IMO, Therefore the ITU could not announce on “Regulation 19” of this convention the use of the AIS as an additional navigational system. The text in verses 22-25 should be should be corrected accordingly.
3. Ship's AIS transmits four, not two groups of data: static, dynamic, voyage related and safety related. For example, mentioned in the article: departure port, arrival port and cargo are voyage related data. The text shall be corrected.
4. It is absolutely necessary to complete the bibliographic reference in verse 38 ([8? ? ]).
5. Figures 4-7 and 11-14 are illegible (font too small) and should be corrected.
Author Response
We thank the reviewer for the insightful comments that have helped us in improving the quality of the manuscript. Please note that the changes are in red-colored font in the revised version.
Point 1: According to the IMO Resolution A.1106(29) AIS is intended to enhance: safety of life at sea and the safety and efficiency of navigation. It shall support ship radar and radar tracking system by assisting in: identification of targets and their navigational status, presentation of targets heading, immediate identification of manoeuvres and more accurate presentation of the targets vectors. As a tool used for this purpose, it can be installed on any convention (AIS class A) or non-convention ship (e.g. AIS class B and AIS receiving device) and it will be difficult to prohibit its installation on any type of vessel. The basic data transmitted by the ship's AIS (ship identification, position, course, speed, navigational status) should be understandable for each AIS, regardless of its class, installed on another ship. Knowing these legal conditions, the authors should, in my opinion, write how the security features proposed in the article will prevent a pirate or terrorist attack carried out from another ship equipped with AIS.
Response to point 1: Indeed, such a discussion is included in our previous work [12]. In order to enhance the self-sustainability of the present paper, we have added appropriate text in the revised version.
Point 2: The SOLAS convention was adopted only by the IMO, Therefore the ITU could not announce on “Regulation 19” of this convention the use of the AIS as an additional navigational system. The text in verses 22-25 should be should be corrected accordingly.
Response to point 2: Agreed, corrected.
Point 3: Ship's AIS transmits four, not two groups of data: static, dynamic, voyage related and safety related. For example, mentionedin the article: departure port, arrival port and cargo are voyage related data. The text shall be corrected.
Response to point 3: Agreed, corrected.
Point 4: It is absolutely necessary to complete the bibliographic reference in verse 38 ([8? ? ]).
Response to point 4: Agreed, corrected.
Point 5: Figures 4-7 and 11-14 are illegible (font too small) and should be corrected.
Response to point 5: Agreed, corrected.
Reviewer 2 Report
I think it is a very good job. I miss, as a suggestion, a comparison with the LRIT: advantages and disadvantages. Also a reference to the general problems of the AIS system: limited scope, not compulsory for all vessels, etc. These observations do not call into question the validity of the work itself.The proposal is entirely feasible and realistic.
Author Response
We thank the reviewer for their insightful comments that helped us in improving the manuscript. Changes made are in red-colored font in the revised version.
Point 1: I think it is a very good job.
Response to Point 1: Thank you for the encouraging comment.
Point 2: I miss, as a suggestion, a comparison with the LRIT: advantages and disadvantages.
Response to Point 2: Agreed. Text has been added to this effect.
Point 3: Also a reference to the general problems of the AIS system: limited scope, not compulsory for all vessels, etc.
Response to Point 3: Agreed. Text has been added to this effect.
Point 4: These observations do not call into question the validity of the work itself. The proposal is entirely feasible and realistic.
Response to Point 4: Thank you for the encouraging comment.